# Neutron Tomography Studies of Two Lamprophyre Dike Samples: 3D Data Analysis for the Characterization of Rock Fabric

**DOI:** 10.3390/jimaging8030080

**Published:** 2022-03-19

**Authors:** Ivan Zel, Bekhzodjon Abdurakhimov, Sergey Kichanov, Olga Lis, Elmira Myrzabekova, Denis Kozlenko, Mannab Tashmetov, Khalbay Ishbaev, Kuatbay Kosbergenov

**Affiliations:** 1Frank Laboratory of Neutron Physics, Joint Institute for Nuclear Research, 141980 Dubna, Russia; zel@jinr.ru (I.Z.); bekhzod@jinr.ru (B.A.); lisa_9477@mail.ru (O.L.); bagdaulet_m@mail.ru (E.M.); denk@nf.jinr.ru (D.K.); 2Institute of Nuclear Physics, Academy of Sciences of the Republic of Uzbekistan, Tashkent 100214, Uzbekistan; mannab@inp.uz; 3Institute of Physics, Kazan Federal University, 420008 Kazan, Russia; 4Institute of Nuclear Physics, Ministry of Energy of the Republic of Kazakhstan, Almaty 050032, Kazakhstan; 5Institute of Geology and Geophysics Named after Kh.M. Abdullaev, State Committee for Geology of the Republic of Uzbekistan, Tashkent 100041, Uzbekistan; halbay@mail.ru (K.I.); qosbergenov93@mail.ru (K.K.)

**Keywords:** neutron imaging, neutron tomography, lamprophyre dike, shape preferred orientations, Raman spectroscopy

## Abstract

The rock fabric of two lamprophyre dike samples from the Koy-Tash granitoid intrusion (Koy-Tash, Jizzakh region, Uzbekistan) has been studied, using the neutron tomography method. We have performed virtual segmentation of the reconstructed 3D model of the tabular igneous intrusion and the corresponding determination of dike margins orientation. Spatial distributions of inclusions in the dike volume, as well as further analysis of size distributions and shape orientations of inclusions, have been obtained. The observed shape preferred orientations of inclusions as evidence of the magma flow-related fabric. The obtained structural data have been discussed in the frame of the models of rigid particle motion and the straining of vesicles in a moving viscous fluid.

## 1. Introduction

Dikes are the main conducting channels of basaltic magma within the Earth’s crust and upper mantle. The flow of melt within such tabular conduit is supposed to be laminar, preserving this state at long distances. Hence, it would be expected, that dikes bear the record of magma flow direction and, therefore, can provide important information about the location and extent of magma sources.

Relatively rapid cooling of moving melted rock facilitates the conservation of imprints of the magma flow, as the development of specific rock fabrics or indicators of the flow: alignment of deformed vesicles, amygdules and ocelli [1,2,3], the crystallographic preferred orientations of minerals [4], shape preferred orientations of tabular phenocrysts [5], surface lineations [6], including wrinkles, finger grooves and anisotropy of magnetic susceptibility (AMS, e.g., [7]). The last-mentioned AMS is one of the important intrinsic indicators of magma movement, characterized by magnetic lineation and foliation. Its development is mostly owed to the alignment of magnetite grains at some imbrication angle, with respect to the flow direction (e.g., [7]). Although numerous works have shown the correlation of measured AMS of dike samples with magma flow direction, there is a manifold in the relation of magnetic ellipsoid axes to the flow direction and dike walls orientation: normal (symmetrical, asymmetrical), intermediate and inverse fabrics (e.g., see an example of a composite lamprophyre dike, studied in [8]). Moreover, in those situations, when the flow fabric was deformed by regional stresses (e.g., [9]) or/and the definitive evidence of flow direction in dikes is lacking, the interpretation of AMS measurements, without additional information extracted from other flow indicators, becomes difficult.

Macroscopic evidence of the flow-related fabric might be found among others in spheroidal (or ovoidal) texture, formed by inclusions of deformed amygdules, vesicles and ocelli (e.g., [3]). However, in most cases the petrofabric analysis of dikes is largely restricted to the flow fabrics exposed on the surface, either of thin sections or of outcrops. From this point of view, X-ray and neutron tomography methods can, potentially, provide a full 3D picture of the spheroidal texture, expressed by spatial distribution, particle morphology and shape orientation distribution of different kinds of inclusions inside the dike volume. It has been shown [10,11,12,13] that neutron radiography and tomography are efficient methods in studying the internal structure of different igneous and metamorphic rocks, as well as construction and archeological materials. In the case of rock material, the high penetration ability of neutrons, as compared to X-ray, allows one to measure rather large volumes, while high contrast between hydrous and anhydrous minerals in neutron images [10,14] provides the necessary conditions for phase segmentation. This is of particular importance, since igneous dikes are relatively dense rocks in the groundmass, composed of amphiboles and feldspars, while the spheroidal inclusions, which may form the flow-related texture, are presented by quartz, feldspars, calcite, and other minerals, rather transparent for neutrons.

In our work, we have examined the internal structure of two samples of lamprophyre dikes from the Koy-Tash granitoid intrusion (Uzbekistan), by means of neutron tomography. These dikes have shown the presence of spheroidal inclusions, which presumably form the texture related to the magma movement. We present an approach to analyze the obtained 3D neutron data of rock samples with dike intrusion and reveal the presence and specifics of 3D spheroidal texture.

## 2. Experimental

### 2.1. Dike Samples Description

The Nuratau region of the South Tien Shan, located at the junction of two continental lithospheric plates (the Kyrgyz-Kazakh and Alay), is characterized by extensive development of magmatic formations. The Koy-Tash granitoid intrusion located in the Nuratau mineralization zone is dissected by numerous dikes of lamprophyres, diorite porphyrites, diabases, quartz monzonite porphyries, granite and granodiorite porphyries of the submeridional direction, totaling more than 30 [15]. Two rock samples with comparable sizes (Figure 1) were taken for tomography study: Sample_1—the sample showing a contact of narrow lamprophyre dike with granodiorite (Figure 1a); Sample_2—a fragment of a wider lamprophyre dike (Figure 1b). Sample_2 is an example of dike sample without evident signs of magma intrusion, except for the wrinkles aligned more or less towards magma movement (Figure 1b). In both rock samples, there are visible spheroidal inclusions against dark dike body, which represent amygdales, ocelli and xenocrysts. Previous microscopic observations of these lamprophyre dikes [15] have shown the predominant content of amphibole (40–70%) and plagioclase (30–60%) and sometimes the presence of quartz (up to 2%) and single grains of decomposed pyroxene.

### 2.2. Raman Spectroscopy

Additional information concerning mineral composition has been obtained by Raman spectroscopy. Raman spectra at ambient temperature were collected using a LabRAM HR spectrometer (HORIBA ABX SAS, Montpellier, France) with the wavelength excitation of 633 nm emitted from He–Ne laser, using 1800 grating, confocal hole of 200 μm, and 50× objective. The measurements were performed at different locations over the surface of the samples. The first group of measurement points corresponds to the dark area of samples (dike groundmass minerals), while in the second group, the points were located on the spheroidal inclusions.

### 2.3. Neutron Tomography

The neutron tomography experiments were carried out at the neutron radiography and tomography facility [16] of the IBR-2 high-flux pulsed reactor (Dubna, Russia). A set of neutron radiography images (projections) has been collected by the detector system based on a high-sensitivity camera with HAMAMATSU CCD chip. In total, 360 radiography projections with a sample rotation step of 0.5° were obtained for each dike sample. The exposure time for one projection was 15 s and the duration of one tomography experiment lasted about 4 h. The imaging data were corrected by the image of dark current in camera and normalized to the image of the incident neutron beam using the ImageJ software [17]. The tomographic reconstruction was performed by SYRMEP Tomo Project (STP) software [18]. Finally, we obtained a large data set containing volume distributions of the neutron attenuation coefficient on the regular 2048 × 2048 × 2048 grid (voxel representation). The size of one voxel in our study is about 98 µm. Mainly due to image blurring, occurring in the conventional pinhole geometry of neutron collimation, the spatial resolution in neutron images was restricted by the minimum size of a resolved item up to ~150 µm. ImageJ software, VGStudio MAX 2.2 software (Volume Graphics, Heidelberg, Germany) and custom Matlab scripts were used in post-processing to visualize and analyze the reconstructed 3D data.

## 3. Results and Discussion

### 3.1. Raman Spectroscopy

Figure 2 shows the selected Raman spectra obtained for both groups of measurement points on the surface of each sample: dike body and inclusions. According to Figure 2a plagioclase is the dominant mineral phase of the feldspathic matrix of Sample_1, although the high level of noise in most of the spectra did not allow us to discriminate amongst different feldspar minerals. In addition, we have found several amygdales, mainly consisting of quartz and sometimes calcite (Figure 2b). Apparently, ocelli and xenocrysts in Sample_1 were represented by plagioclase and alkali feldspars, as inferred from Raman spectra in Figure 2b.

The set of Raman spectra of dike Sample_2 is similar to those collected for dike Sample_1 (Figure 2c,d). The Raman spectra of Sample_2 confirm the presence of the feldspathic matrix, composed of plagioclase and alkali feldspars (Figure 2c). The qualitative distinction between feldspars was drawn on the basis of their major Raman bands, as listed in [19]. The spheroidal inclusions (amygdules, ocelli and possibly xenocrysts) in Sample_2 mostly contained quartz and plagioclase (Figure 2d).

The phase of fine-grained amphibole was not distinguished in the obtained spectra due to high photoluminescence and rough surfaces of both dike samples. Nevertheless, the results of Raman spectroscopy are in agreement with microscopic observations, showing the major contribution from feldspars (Figure 2a,c).

### 3.2. Neutron Tomography

3D models of dike Sample_1 and Sample_2, reconstructed from tomography data, are presented in Figure 3. The corresponding 3D data sets were deliberately converted to a 16-bit format (integers 0…65,535, see also the color bars in Figure 3) for memory saving while post-processing. In Sample_1, we have clearly seen two different sub-volumes (Figure 3a,c). The dike body is highly attenuative for neutrons (amphibole rich sub-volume), which dissects the relatively transparent granodiorite host, composed mainly of feldspars and quartz. The boundaries between these sub-volumes are almost parallel, although their orientation does not coincide with the reference coordinate frame. 

In both samples, there are rounded regions within the dike body with high attenuation of neutrons, apparently corresponding to hydrous phenocrysts, either of amphibole or mica, or of both. According to the Raman spectroscopy results (Figure 2b,d), the regions with low attenuation of neutrons represent the spheroidal inclusions, containing mostly feldspar and quartz. These low attenuative inclusions are well distinguished in 2D slices (Figure 3c,d). In Sample_1, they have shown a tendency to align along the direction parallel to the dike walls, as seen in Figure 3c.

For further analysis of the tomography data of Sample_1, the following tasks were required to be fulfilled: (1) to separate the dike body from granodiorite host and eliminate its influence on phase segmentation; (2) to determine orientation of the dike walls and segment the planes in 3D, along which the magma flow took place. Due to image blurring and non-homogeneous phase distribution, the standard algorithms to segment and clip the corresponding volumes did not bring satisfactory results. We have applied our original algorithm to separate the dike volume and dike walls segmentation. This approach is presented in more detail below.

### 3.3. 3D Data Analysis: Dike Walls Orientation and Segmentation

The plane parallel contact of the dike with granodiorite in Sample_1 resembles the presence of layering: the layer of the dike is in between the layers of granodiorite. Such 3D structures can be analyzed using the scanning method [12], designed to reveal the foliation and lineation fabrics in 3D images. The scanning method is based on the continuous scanning of 3D data at different angles of view and subsequent calculations of the variation of intensity values (degree of heterogeneity), obtained on the plane or line elements at each angle. Following the procedure fully described in [12], the 3D data have been rotated with a 5-degree step so that the z-axis of the data set moved over the upper hemisphere (2π coverage). At each rotation step, the 3D data have been divided into the plane elements (slices), perpendicular to the current z-axis and line elements, which are parallel to the current z-axis. Two quantities are then calculated: in-plane variation $\sigma_{P}$—weighted standard deviation of the image intensity collected over plane elements; in-line variation $\sigma_{L}$—weighted standard deviation of the image intensity collected over line elements. The resulted distributions of $\sigma_{P}$ and $\sigma_{L}$ on a unit sphere can reveal certain planes or directions (foliation or lineation), related to the structural ordering within the sample volume. 

Figure 4a,b demonstrates the obtained spatial distributions of $\sigma_{P}$ and $\sigma_{L}$ plotted on the stereographic projection. Both distributions are mutually consistent and show the presence of layering or foliation. In the $\sigma_{P}$ diagram (Figure 4a), there is a pole of maximum values around a certain direction—normal to the layers or dike walls. In the $\sigma_{L}$ diagram (Figure 4b), there is a belt of maximum values lying within a certain plane—the plane of dike walls. Thereby, from $\sigma_{P}$ and $\sigma_{L}$ diagrams, we have unambiguously determined the orientation of the dike walls by identifying the orientation of the normal.

To determine the spatial position of the dike walls in Sample_1, the 3D data were brought to the orientation where the z-axis was parallel to the normal of the dike walls, according to Figure 4a,b. At this orientation, the dike walls were in a horizontal plane. The segmentation of the dike body boundaries was then equivalent to the determination of the specific slice numbers. We found the slice numbers corresponding to the dike walls by choosing an appropriate threshold for the intensity values averaged over horizontal slices. For 3D visualization, the model of dike walls was built from the mask images formed after the thresholding operation.

In Figure 4c,d virtual models of dike walls dissecting Sample_1 are shown. From these figures, it is clearly seen that the dike walls, determined semi-automatically, are in good agreement with an actual variation of the neutron attenuation coefficient, caused by the igneous intrusion. Although the boundaries between dike body and granodiorite host are not perfectly flat, they are practically parallel, preserving this property across the overall volume. The obtained model of the dike walls allowed us to unambiguously separate the dike body from the granodiorite.

### 3.4. 3D Data Analysis: Size Distribution and Shape Orientations of Inclusions

Phase segmentation of spheroidal inclusions was performed by thresholding the already separated dike volume of Sample_1 and 3D model of Sample_2. Each of the segmented vesicles was assigned a serial number (label) in the image to perform the morphological analysis of each particle. In Figure 5 the resulted spatial and size distributions of inclusions in dike volumes are shown. The sizes of inclusions were evaluated using their equivalent diameter—the diameter of a sphere with a volume equal to the volume of inclusion. In both samples, the inclusions are distributed almost uniformly, without any dependence on their size. According to the size histograms (Figure 5), most of the inclusions do not exceed 5 mm and the largest of them are about 10 mm. The presence of some alignment of vesicles is seen in Sample_1 (Figure 5a), while in Sample_2, there is no such clear evidence (Figure 5b).

To quantify the alignment of inclusions, one needs to mathematically define the shape orientation itself. We have used the Legendre ellipsoid approximation [11,12], where each particle is approximated by the ellipsoid, which has exactly the same inertia as the particle has. Thereby, we defined the shape orientation as orientation of three principal axes of Legendre ellipsoid or equivalent of the inertia moment tensor. The components of the inertia moment tensor were calculated for each of the labeled inclusions and the corresponding principal axes of inertia (corresponding to the eigenvalues *I*_min_, *I*_int_, *I*_max_) were used as orthonormal axes of the shape of these particles. The obtained stereoplots of the spatial distributions of principal axes for the inclusions’ shape are shown in Figure 6. For these graphs, we have taken only those particles whose Legendre’s ellipsoid dimensions differ by at least 1.5 times and equivalent diameter is more than 0.5 mm. It has been done to exclude small particles or those that are weakly elongated or flattened, whose orientation may be ambiguously determined.

The axis orientations of inclusions (Figure 6) show a non-uniform distribution and this is especially seen for the smallest axes with the highest inertia *I*_max_. Kamb contours [20], applied to highlight the presence of preferred orientations in the axis distributions, statistically significant. On these stereoplots, there is a pole with the highest concentration of Imax axes in the direction sub-parallel to the laboratory xy-plane in Sample_1, and in the direction sub-parallel to the laboratory z-axis in Sample_2. Other axes demonstrate less localization. In Sample_1, the axes of minimum inertia form a belt, perpendicular to the pole in the corresponding *I*_max_ diagram (Figure 6a). The axes of intermediate inertia values form a part of the belt in the same plane as for *I*_min_ axes. In Sample_2, the axes of the minimum and intermediate inertia values have poles in one plane but in mutually perpendicular directions.

To provide a more confident evaluation of the shape’s preferred orientations of inclusions, we have performed the non-parametric bootstrapping of inertia tensor data sets. It is similar to the procedure commonly used in the magnetic anisotropy analysis [21]. In bootstrap calculations, we used normalized inertia tensors to minimize the influence of particle sizes and performed, overall, 10,000 simulations. Confidence regions were calculated for the orientation distance (e.g., [22]) between the orientation of the mean inertia tensor and the corresponding orientation of inclusions.

The distributions of principal axes of the mean inertia tensors, obtained after the simulations, are shown in Figure 7. For both samples, the minimum spread out has been observed for the maximum inertia axis. Intermediate and maximum inertia axes in Sample_1 form a belt (Figure 7a), while in Sample_2, these axes are separated on the stereoplot (Figure 7b). For comparison, in Figure 7a, the orientation of the dike walls are shown for Sample_1. Intermediate and maximum inertia axes of the mean inertia tensor are almost parallel to the dike walls. Moreover, the projection of the dike walls and its normal lie within 95% confidence regions of particle orientations (Figure 7a). The angle between the minimum inertia axis of the mean inertia tensor and normal to the plane of the dike walls is 7°, with a maximum deviation of about 30° within 95% confidence region. This means that inclusions tend to align their smallest axis perpendicular to the dike walls and, hence, to the magma flow direction. In the dike of Sample_2 the intermediate and maximum inertia axes are oriented subparallel to the *xy*-plane (Figure 7b), following the orientation of wrinkles observed on the sample surface (Figure 1b).

### 3.5. Discussion

The observed shape preferred orientations in solidified magma are closely related to the fluid dynamics of melted rock. Magma movement within a dike can be approximated by the models of moving Newtonian or Bingman fluid in a tabular conduit (e.g., [23]). The velocity gradient appeared due to friction of the fluid with the conduit’s walls, which defines the shear stresses that act on all elements or inclusions in the fluid: solid (phenocrysts, xenocrysts), immiscible liquid or gaseous vesicles. The common tendency of inclusions to align subparallel to the flow direction of fluid has been widely observed in dikes, as well as predicted by laboratory and theoretical models [24,25,26,27], although in most cases, the cyclic motion is implied, depending on the shear strain and instability of motion in the case of non-axisymmetric particles [28].

In Sample_1, inclusions in the dike are predominantly represented by amygdales, ocelli and xenocrysts of the prolate and oblate shapes. This mixture of differently shaped inclusions shares a common fabric, with a clearly defined foliation plane, inclined on the average by 7° to the dike walls and the smallest axes almost perpendicular to the flow direction (Figure 6 and Figure 7). Although the largest axes of inclusions are subparallel to the flow direction due to shear stresses, the observed bulk configuration of preferred orientations of principal axes does not coincide with the models of motion of rigid axis-symmetric spheroids, but only with the tiled imbrication model [29]. In rigid particles models (e.g., [28]), the stable states are those when the smallest axis is oriented perpendicular to the plane of velocity gradient, during the tumbling mode for ideally prolate spheroids (rods) and log-rolling mode for ideally oblate spheroids (disks). However, it should be emphasized that rotations of triaxial ellipsoid around its symmetry axes, under the conditions of shear flow, are not stable, contrary to the cases of axisymmetric spheroids [28].

A similar comparison can be done for the case of deforming initially spherical gaseous or liquid vesicles in the shear flow, when the largest axes of the strained ellipsoids are subparallel to the flow direction with an imbrication angle, depending on the capillary number [26,27]. It has been shown [27] that under the simple shear flow, a spherical vesicle deforms into elongated triaxial ellipsoid. The largest axis of the vesicle increases with strain growing, while other axes are decreasing. It is important to mention that the smallest axis (axis of *I*_max_) is inclined to the normal of the conduit walls, as the largest axis (axis of *I*_min_) does to the flow direction [27,30]. Such orientation of the deformed vesicle may explain the preferred orientations of inclusions that we observed in Sample_1 (Figure 6a and Figure 7a). Apparently, most of the inclusions in the moving magma were presented by immiscible liquid droplets and gaseous vesicles, which later, were crystallized or filled with secondary minerals and formed ocelli and amygdales.

In our work, Sample_1 represents a reference sample with known orientation of the magma flow plane, while Sample_2 is a typical example, when primary information about the direction of the magma movement is lacking. From neutron tomography measurements and 3D data analysis of the reconstructed sample model, we have obtained clearly defined preferred orientations of inclusions in Sample_2 (Figure 6 and Figure 7). Due to the common tendency of inclusions to align their largest axes subparallel to the flow direction, the spatial distribution of *I*_min_ axes may provide a reasonable estimation of the magma flow direction. 

The 3D shape fabric of different inclusions in solidified magma, obtained from neutron tomography measurements, may provide valuable information, not only about the direction of magma ascent, but may be also helpful in understanding the relation between fluid dynamics and the motion of different inclusions or their straining. The high penetration ability of neutrons into rather dense rocks and the application of 3D structure analysis may improve the quantitative description of inclusions’ morphology and orientation (including the improvement of statistics) in studying the influence of the type of flow and shear rates on the inclusions in magmas (e.g., in obsidians [30]).

## 4. Conclusions

Neutron tomography measurements of two lamprophyre dike samples were performed, aiming to reveal the flow fabric of moving magma in solidified rocks. The high penetration ability of neutrons and large difference in neutron attenuation in hydrous and an-hydrous minerals provided the opportunity to obtain a spatial distribution of different inclusions, as amygdules, ocelli, xenocrysts inside the dikes. Within the analysis of 3D neutron data, we have shown the segmentation of tabular igneous intrusion and determination of its orientation, as well as determination of 3D distribution of inclusions, their size distribution and orientations of their inertia axes. In both studied samples, inclusions have the shape preferred orientations, related to the motion of inclusions during the magma movement. The observed shape fabric has been discussed in the frame of models of motion of rigid particles and the straining of vesicles in the fluid flow. We consider that the presented neutron tomography results and algorithms of neutron data analysis have shown a valuable contribution of neutron tomography to the petrofabric studies of dikes.

## Figures and Tables

**Figure 1 jimaging-08-00080-f001:**
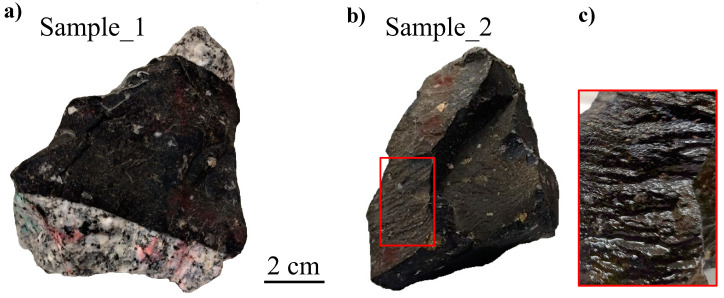
Photographs of the studied lamprophyre dike samples Sample_1 (**a**) and Sample_2 (**b**). Several flow-related wrinkles are highlighted by red color. The enlarged region of wrinkles is shown (**c**).

**Figure 2 jimaging-08-00080-f002:**
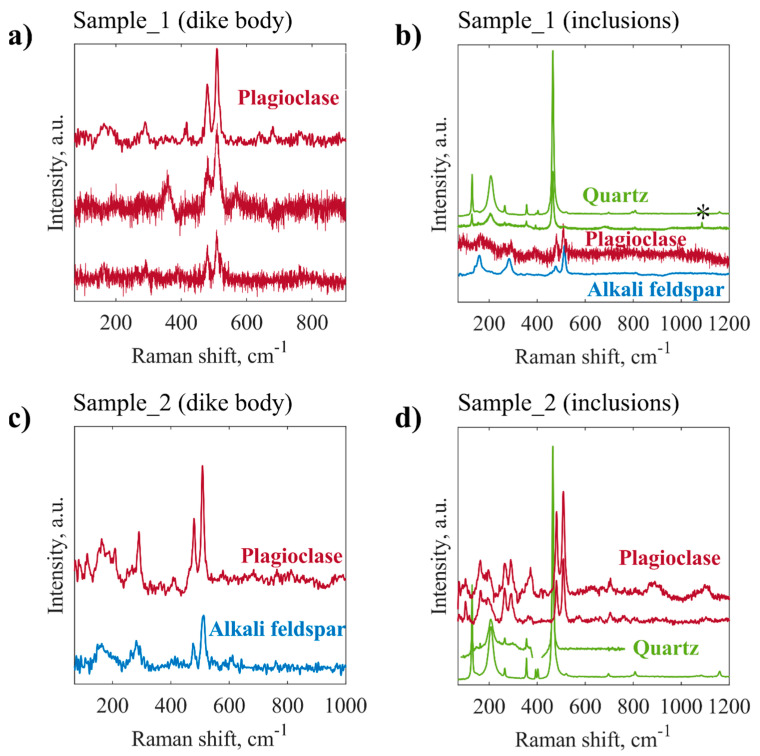
The selected Raman spectra obtained from different points on the surfaces of the studied rock samples, Sample_1 (**a**—dike body, **b**—inclusions) and Sample_2 (**c**—dike body, **d**—inclusions). Asterisk in panel (**b**) corresponds to the calcite principal Raman shift of 1086 cm^−1^.

**Figure 3 jimaging-08-00080-f003:**
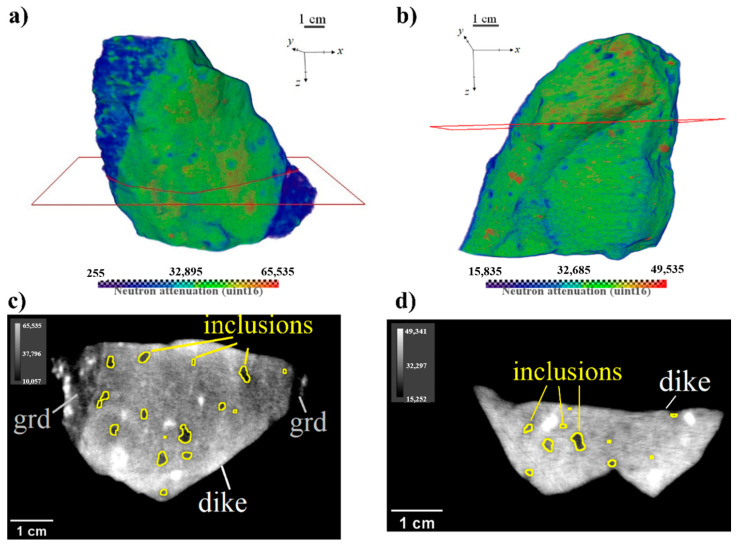
Reconstructed virtual models of the studied dike samples: the upper row—3D models; the lower row—selected 2D slices (‘grd’—granodiorite). (**a**,**c**)—Sample_1; (**b**,**d**)—Sample_2. The spatial positions of the slices on the three-dimensional model are marked. Color bars show the variation of neutron attenuation coefficient values converted to 16-bit integers.

**Figure 4 jimaging-08-00080-f004:**
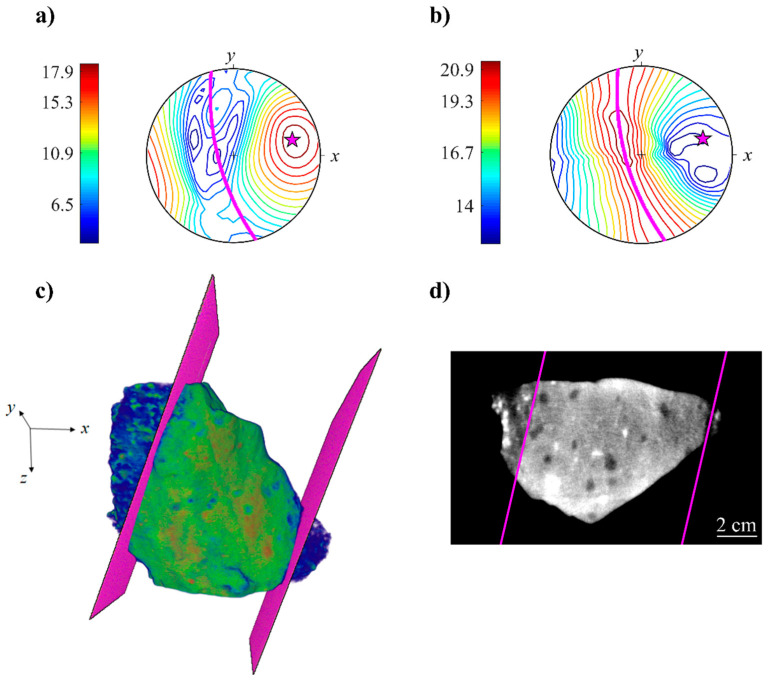
The calculated stereoplots of in-plane $\sigma_{P}$ (**a**) and in-line $\sigma_{L}$ (**b**) spatial variations of neutron attenuation coefficient for Sample_1. The scale bar in the diagram is shown in a.u. Magenta stars and lines denote the direction of maximum $\sigma_{P}$ and the corresponding plane normal to this direction. The obtained virtual models of plane-parallel dike walls are shown by magenta color in 3D model (**c**) and selected 2D slice (**d**).

**Figure 5 jimaging-08-00080-f005:**
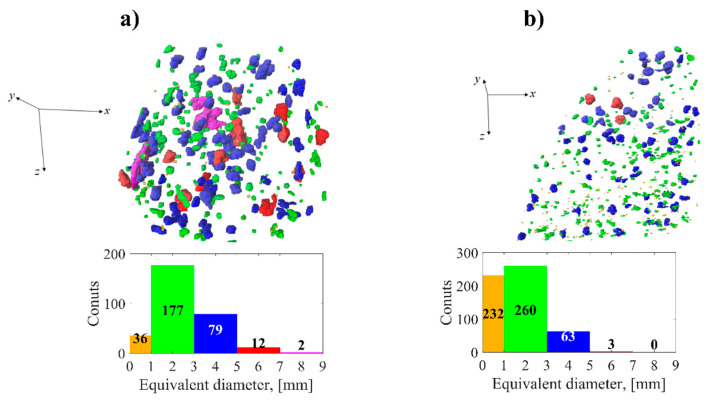
Spatial and size distributions of the segmented inclusions in Sample_1 (**a**) and Sample_2 (**b**). Particles in the 3D models are colored according to the histograms below.

**Figure 6 jimaging-08-00080-f006:**
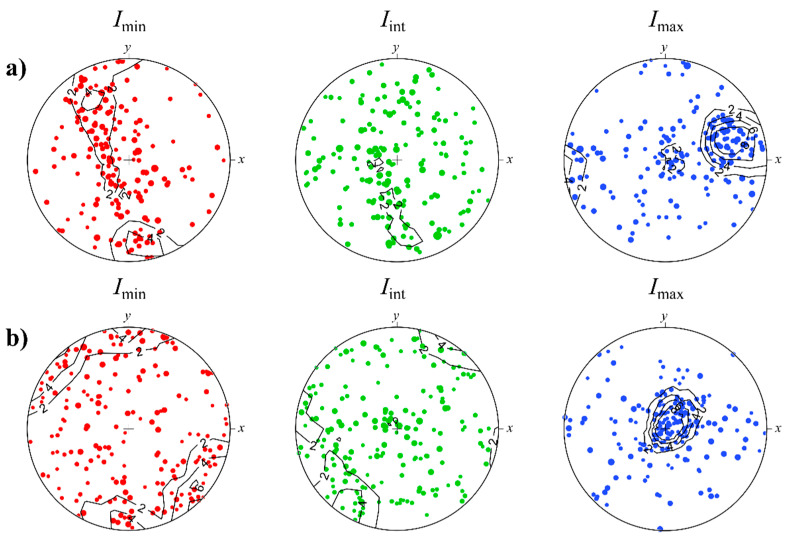
The upper hemisphere stereographic projections of principal axes of inertia tensor of inclusions in Sample_1 (**a**) and Sample_2 (**b**). The size of symbols is proportional to the equivalent diameter of particles. Kamb contours (E = 3σ) are shown at 2σ, 4σ, 6σ, 8σ levels.

**Figure 7 jimaging-08-00080-f007:**
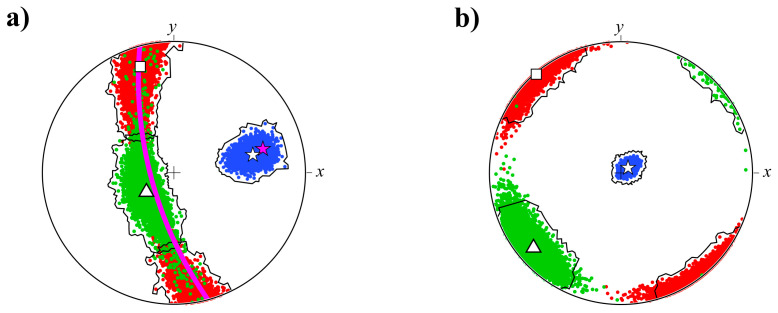
The upper hemisphere stereographic projections of principal axes of the mean inertia tensor of vesicles in Sample_1 (**a**) and Sample_2 (**b**) after non-parametric bootstrapping (see text for details). Directions of the principal axes are colored in the same way as in Figure 6: red—*I*_min_, green—*I*_int_, blue—*I*_max_. White stars denote the orientation of *I*_max_ principal axis of the mean inertia tensor. Magenta stars and lines denote the direction normal to the plane of dike walls and the plane itself correspondingly. Black lines contour the 95% confidence regions.

## Data Availability

No new data were created or analyzed in this study. Data sharing is not applicable to this article.

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
