# Peer review of "Neutron Tomography Studies of Two Lamprophyre Dike Samples: 3D Data Analysis for the Characterization of Rock Fabric"

_2313-433X, 2022, doi:10.3390/jimaging8030080_

Round 1

Reviewer 1 Report

I ave only small remarks which could be found on the attached pdf. file.

My recommendation: Accept after minor revisons

Author Response

Dear Reviewers,

We would like to re-submit our paper

" Neutron Tomography Studies of Two Lamprophyre Dike Samples: 3D Data Analysis for the Characterization of Rock Fabric "

 by Ivan Zel, Bekhzodjon Abdurakhimov, Sergey Kichanov, Olga Lis, Elmira Myrzabekova, Denis Kozlenko, Mannab Tashmetov, Khalbay Ishbaev and Kuatbay Kosbergenov

 Ref.:  jimaging-1617568 

We would like to sincerely thank Reviewers for careful reading of the manuscript and providing the useful remarks and comments.

We had make following explanations and corresponded corrections:

A small note. I would advise authors to add their ORCID ID numbers.

Thanks for the comment. But we have a large team and not everyone has an ORCID. Moreover, according to the requirements of the journal, this is not required. However, the correspondent author's ORCID was cited.

Please reformulate «Spatial distributions of inclusions (amygdules, ocelli, xenocrysts) within the dike volume have been obtained as well as further analyzed size distributions and shape orientations of inclusions.»

We had done this.

The images are not of the best quality. I would suggest to edit them using an appropriate software. Also I would ask to increase their size so that to make the details better evidenced.

We did it. We change the Figure.

Also, presented an enlarged image of the Sample_2 area where wrinkles are evident (the encircled area)Please mark by a rectangle the position of images c and d on the tomographic images a and b.

We did it. The Figure content was changed. The Figure’s captions are changed also.

Please include the images in the text where they were for the first time cited. This remark is valid for all figures.

We change places of Figures.

Please give more details on how this was done!

All this terminology, procedures and designations were made in accordance with the publication [Zel and et al, Tectonophysics 2021, 812, 228925.]. If we add extended description of all the procedures, then this will cause excessive quoting of those already described in this publication.

Sincerely yours,

Kichanov Sergey,

on behalf of co-authors

Reviewer 2 Report

This is a very impressive study of inclusions and grains in magmatic rock using neutron computed tomography. I lack the background in geology and also in mathematics for the 3D analysis presented here to thoroughly question these parts, but the authors have presented their case very well, and the details sound very plausible to an outsider of the field like me.

The paper is written in excellent English.

It should be accepted in its present form, but I would still like to give some advice for the future in my own field of expertise, neutron tomography. The number of angular projections chosen (360 over 180 degrees) is way too low for a good reconstruction from 2048x2048 pixel images. According to Kak & Slaney ("The mathematics of computerized tomography", available online for free), one would ideally need Pi/2 as many projections as there are pixels in one line, so about 2,800 , but that sharpness curve is somehow asymptotic, influenced by more sources of image unsharpness and blur.

We found about 900 projections for a 2048x2048 pixel camera to be a good compromise between image sharpness and measurement time; using 1200 projections leads to a still noticeable improvement in image sharpness, but this is not worth the extra measurement time.

So if the authors choose more projections next time, they will find improvement in the sharpness of boundaries between grains and inclusions, and the surrounding matrix, and they may save some time and effort in the very complex refined segmentation process they described so well in this paper.

Thanks for the great paper, and good luck for future measurements!

Author Response

Dear Reviewers,

We would like to re-submit our paper

" Neutron Tomography Studies of Two Lamprophyre Dike Samples: 3D Data Analysis for the Characterization of Rock Fabric "

by Ivan Zel, Bekhzodjon Abdurakhimov, Sergey Kichanov, Olga Lis, Elmira Myrzabekova, Denis Kozlenko, Mannab Tashmetov, Khalbay Ishbaev and Kuatbay Kosbergenov

Ref.:  jimaging-1617568 

We would like to sincerely thank Reviewers for careful reading of the manuscript and providing the useful remarks and comments.

This is a very impressive study of inclusions and grains in magmatic rock using neutron computed tomography. I lack the background in geology and also in mathematics for the 3D analysis presented here to thoroughly question these parts, but the authors have presented their case very well, and the details sound very plausible to an outsider of the field like me.

The paper is written in excellent English.

Thank you so much for the good words addressed to our manuscript. Thank you for your careful reading and for your comments.

It should be accepted in its present form, but I would still like to give some advice for the future in my own field of expertise, neutron tomography. The number of angular projections chosen (360 over 180 degrees) is way too low for a good reconstruction from 2048x2048 pixel images. According to Kak & Slaney ("The mathematics of computerized tomography", available online for free), one would ideally need Pi/2 as many projections as there are pixels in one line, so about 2,800 , but that sharpness curve is somehow asymptotic, influenced by more sources of image unsharpness and blur.

We found about 900 projections for a 2048x2048 pixel camera to be a good compromise between image sharpness and measurement time; using 1200 projections leads to a still noticeable improvement in image sharpness, but this is not worth the extra measurement time.

So if the authors choose more projections next time, they will find improvement in the sharpness of boundaries between grains and inclusions, and the surrounding matrix, and they may save some time and effort in the very complex refined segmentation process they described so well in this paper.

We fully agree with your remark about the number of projections in the neutron tomography experiment. However, there are several points for the possibility of using a limited number of projections in our experiment:

Firstly, in neutron tomography experiments, we have a hard condition for the radiation background. So, that increasing the experiment time by 2-3 times reduces the service life of the detector at the same time.

Secondly, in our manuscript, we used the SIRT reconstruction method of tomography reconstruction, which is positioned precisely to reduce the number of projections during reconstruction procedure (Ex: D. Micieli, Scientific Reports 9, 2450 (2019)). 

And the last circumstance, for our studies of the orientation of grains and inclusions, an increase in the accuracy of determining the surface boundary will not increase (some noticeably) the accuracy of determining the orientation of the grain in the approximation that we used.

But in any case, your remark is correct and very useful.

Thanks for the great paper, and good luck for future measurements!

Thank you so much for your kind wishes.

Sincerely yours,

Kichanov Sergey,

on behalf of co-authors
